# An Idealized Clinicogenomic Registry to Engage Underrepresented Populations Using Innovative Technology

**DOI:** 10.3390/jpm12050713

**Published:** 2022-04-29

**Authors:** Patrick Silva, Deborah Vollmer Dahlke, Matthew Lee Smith, Wendy Charles, Jorge Gomez, Marcia G. Ory, Kenneth S. Ramos

**Affiliations:** 1Health Science Center, Texas A&M University, 8441 Riverside Pkwy, Bryan, TX 77807, USA; jorgejgomez@tamu.edu (J.G.); kramos@tamu.edu (K.S.R.); 2School of Public Health, Texas A&M Health Science Center, 212 Adriance Lab Rd., College Station, TX 77843, USA; deborahvd@gmail.com (D.V.D.); matthew.smith@tamu.edu (M.L.S.); mory@tamu.edu (M.G.O.); 3BurstIQ, 9635 Maroon Circle, #310, Englewood, CO 80112, USA; wendy.charles@burstiq.com

**Keywords:** genomic, registry, chronic disease, health disparity, electronic medical record, cancer

## Abstract

Current best practices in tumor registries provide a glimpse into a limited time frame over the natural history of disease, usually a narrow window around diagnosis and biopsy. This creates challenges meeting public health and healthcare reimbursement policies that increasingly require robust documentation of long-term clinical trajectories, quality of life, and health economics outcomes. These challenges are amplified for underrepresented minority (URM) and other disadvantaged populations, who tend to view the institution of clinical research with skepticism. Participation gaps leave such populations underrepresented in clinical research and, importantly, in policy decisions about treatment choices and reimbursement, thus further augmenting health, social, and economic disparities. Cloud computing, mobile computing, digital ledgers, tokenization, and artificial intelligence technologies are powerful tools that promise to enhance longitudinal patient engagement across the natural history of disease. These tools also promise to enhance engagement by giving participants agency over their data and addressing a major impediment to research participation. This will only occur if these tools are available for use with all patients. Distributed ledger technologies (specifically blockchain) converge these tools and offer a significant element of trust that can be used to engage URM populations more substantively in clinical research. This is a crucial step toward linking composite cohorts for training and optimization of the artificial intelligence tools for enhancing public health in the future. The parameters of an idealized clinical genomic registry are presented.

## 1. Introduction

Precision medicine and health disparities are inextricably linked [1], as existing disparities are amplified in oncology where access to, and the use of, molecular testing remains limited [2,3]. Consequently, knowledge about the distribution and clinical actionability of certain molecular pathologies remains poorly characterized in underrepresented populations [4]. These disparities are well documented and reviewed thoroughly elsewhere [5]. For example, in pharmacogenomics, the gray area between variants of unknown significance and clinically actionable genetic variants tends to be occupied by rarer and emergent variants [6]. Progress toward the clinical interpretation and actionability of emergent variants has been impeded because adequate documentation of the relevant clinical context is at a nascent stage. The underrepresentation of populations harboring rarer variants in research (the Preferred Cohort effect, [7,8]) has limited the availability of the clinical outcome data and mechanistic insights necessary to define clinically actionable pathways and guidelines for those variants. The absence of such characterization can also limit referrals of patients to interventional trials, where the enrollment criteria for experimental therapeutics in oncology increasingly requires the appropriate molecular characterization of disease. This, in turn, limits the commercialization and clinical validation of biomarker assays and clinical decision tools [9].

An example is the NCI-MATCH program, a cross-cutting and paradigm-shifting interventional basket trial master protocol with over 5000 subjects enrolled at over 1000 sites [10]. While the MATCH protocol presents a noble and potentially transformative oncology study strategy, the protocol does not comprehensively account for the longitudinal and health economic outcome measures that could catalyze changes in reimbursement and clinical practice that further perpetuate disparities. The implementation of tests to evaluate well-established oncogenes in managing individuals at risk for hereditary ovarian and breast cancers [2] and Lynch Syndrome [11] illustrate the disparities inherent in engaging underrepresented populations. Mortality in black women with hereditary breast cancers is 42% higher than in non-Hispanic white women [12], despite widespread reimbursement of BRCA testing, suggesting there is more at work than simply access to care/insurance coverage. In oncology, clinicogenomic protocols are widely used and drive the case for evidence-based practice [13]. The term “clinicogenomic” refers to datasets combining the genetic (and often other -omic bioanalyte) characteristics of disease, coupled with the phenotypic annotation of clinical outcomes. In pharmacogenomics, documenting the linkages between genotypes, practice guidelines, and population health remains challenging [14]. The nexus of oncology and pharmacogenomics is thus: chemotherapeutic toxicities and long-term clinical outcomes have been observed to differ substantially among populations [15]. Validating the mechanistic and pharmacogenomic basis of these differences remains elusive because of the lack of participation in clinical studies and related healthcare disparities among certain populations.

The first appearance of the clinicogenomic terminology in the literature was in 2003. It applied to the development of predictive tree models to partition breast cancer patients into increasingly homogeneous subgroups based on clinical outcomes and gene expression profiles in tumors from a prospectively enrolled cohort [16]. More recently, clinicogenomic databases have reached population scale (vs. cohort scale) and include real world data from electronic medical records [17], creating opportunities to train artificial intelligence models to create synthetic controls [18], external control arms [18], and digital twins [19,20]. Conducting a prospective clinical oncology study, whether interventional or observational, is challenging and costly in the best oncology specialty practices and in academic medical centers, which are optimized for robust clinical research. Doing so in less optimal oncology centers (e.g., in community health, primary care, and rural areas) is challenged by austere environments with scant clinical research infrastructure, lack of committed staffing, limited data management infrastructure, and lack of appropriate expertise. The infrastructure for cancer clinical trials has been neglected in forward regions of the healthcare system (i.e., primary care, long-term care) [3], with only a few examples of large cohorts of multiethnic populations [21]. Any effort to address health disparities and underrepresentation in clinical oncology research would be well served by a plan to holistically incorporate biorepository, registry, and longitudinal outcomes (including health economic outcomes) to optimally leverage the participation of, and benefits accruing to, underrepresented groups. An idealized clinicogenomic registry unlocks potential virtualized clinical trial and public health approaches to engage underserved and underrepresented populations in the spirit of patient-centricity, equipoise, justice, and benevolence.

## 2. Barriers and Benefits of an Idealized Clinicogenomic Registry

### 2.1. Underlying Cultural and Social Determinants of Health (SDOH)

The forum where the information is collected can also impact transparency and willingness to disclose. For minorities and other marginalized populations, trust is also an especially meaningful factor in participation and engagement rates and is related to many of the identified barriers [22,23,24,25]. In rural populations, a trusted primary care provider might be better positioned to enroll a subject in a longitudinal registry program versus a specialist at an urban academic medical center to whom the patient has recently been referred and with whom they do not yet have a personal relationship. The reality is that robust infrastructure for registries and biorepositories exists in specialties within academic medical centers whose business model is based on referral to these specialties, and the access to innovative care that such an environment provides. Our view is that this reality is at odds with the relationship and trust-based approach necessary to engage disadvantaged rural populations in registries and longitudinal clinical research. For impoverished and rural populations, we propose that technology design and transparency are important determinants of successful patient-centered engagement, data management, and eventual adoption of a curated case-level health data ecosystem.

Recently, patients and research participants from underserved populations have begun to advocate for property rights in, and even compensation for, the use of their health data provided in the course of research [26,27]. Mikk et al. argue that a patient more actively engaged with their data is likely to result in personal and social benefits in the form of better adherence and outcomes [28]. However, the role of patient ownership or control of the use of their data is problematic, as the legal precedent does not currently provide for complete patient property rights in clinical data [29,30]. It is argued that such blanket rights would have social costs in the form of impeding research and healthcare [29]. A clarification of the uses (vs. the binary question of ownership or co-ownership) of patient data is perhaps a practical middle ground to this emergent debate, but these issues are not well addressed in the academic literature [31]. Shifting debate to the degree of direct control by the patient and the transparency of that use for the patient has the potential to shape policy in a way that better addresses the tradeoffs between the social benefits and benefits to participants. In fact, patient advocates have floated the idea of compensating patients for the use of their data [32]. Regrettably, this is impractical because of the nonfungible nature of most types of medical data. As a practical matter, clinical data are much like research, where the economic value is not readily appropriable to the individual contributors to that social good [33,34]. The ideal clinicogenomic registry would more directly link the research use of data and patient benefits such that policy on clinical data use is afforded a patient-centric approach.

### 2.2. Institutional Review Board (IRB) and Protocol Adherence

Most administrative processes at the interface between the IRB, stakeholders, and investigators are analog. Indeed, protocols and associated documents are distributed in digital form, and compliance is tracked in digital systems meeting statutory and institutional security and privacy specifications. However, two pillars of clinical research ethics remain highly analog and, thus, non-dynamic: (1) IRB review of the protocol; and (2) protocol amendment. Data governance processes that enable multistakeholder governance with dynamic, granular, and digital tools have been slow to emerge. Specifically, data governance in many clinical trials, and especially academic clinical trials, happens through static data use agreements and consent/authorization forms, despite the trend toward use of electronic data capture and the use of cloud computing in clinical research. For example, in a registry or clinical trial, different categories of clinical data (individually identifiable information, limited datasets, health information, de-identified data, and third-party data) often warrant differential regulatory, ethical, and legal treatment. The movement of regulated data (i.e., any data with contractual or protocol constraints on sharing or use) between patient/participant, investigator, and research end users is often not reflected in a digital ledger. The examination of data traffic by ethical stewards of a study (i.e., IRBs, privacy officers) typically only arises if provoked by questions of protocol adherence. Audits are inefficient and laborious. Analog-informed consent/authorization forms, data use agreements, and protocols are not often readily amenable to enabling these distinctions in workflows. This leaves patients, researchers, and institutional stewards trapped and constrained in a static set of rules for engagement. Data structures, data standards, and data indexing all serve to clarify these issues through protocol development and ethical review. Still, it is increasingly important that these project features be dynamic and, thus, increasingly digital [35]. Administrative processes must allow dynamic, digital, and multimedia content, which is at odds with business processes in many organizations that are paper-document driven (as seen with the transition from wet-ink signing to DocuSign) [36]. If administrative processes are not digital, they are increasingly unavailable and opaque to governance stakeholders, including human subjects protection offices (HSPO) and patients themselves.

A dizzying array of technological, legal, and ethical considerations challenge HSPOs and IRBs that review protocols involving registries, biobanking, and longitudinal clinical data curation [37]. An explosion in medical data, and new uses of it, continue to amplify the ethical complexity of clinical research. The zero-sum notion of data ownership itself can impede clinical research at nearly every administrative unit in an institution. Liddell et al. [38] argue that the propertization of health data likely does little to enhance patients’ self-determination. Indeed, if many health systems have difficulty using their troves of clinical data as currency, what likelihood does a patient have in doing so fruitfully? In our experience, these challenges are further compounded by investigators’ natural lack of knowledge about emergent issues at this nexus of technology, ethics, and legal/administration. This complexity is a recipe for time-consuming machinations and high transaction costs. Few IRBs and few investigators possess the expertise to navigate fit-for-purpose approaches that align with the research requirements and the many data stakeholders inherent in a longitudinal research protocol or registry. The participation of research subjects from underrepresented populations in the IRB review process is equally important to address cultural competence, the appropriateness of the informed consent, and the overall soundness of studies that might generate data in special populations. The ideal clinicogenomic registry replaces the ownership quandary with the concept of a participant-centric governance framework for permitted data uses, around which all stakeholders can align dynamically in real time.

### 2.3. Informed Consent

In response to unethical human experimentation, the Tuskegee Syphilis Study, the US Congress commissioned a principled analytical framework to “guide the resolution of ethical problems arising from research involving human subjects” [39]. Three principles of informed consent are articulated in the Belmont Report, the result of the Congressional investigation: information, comprehension, and voluntariness. Another school of thought is to only allow research if the patient consents to a specific line of study or research use, or if the IRB issues waivers of informed consent and authorization in accordance with the Office for Human Research Protections and the Health Insurance Portability and Accountability Act of 1996 (HIPAA), respectively [40]. The scope of consent and HIPAA authorization for future unspecified research, or allowing the interrogation of specimens or the retroactive analysis of identifiable personal data, remains controversial. The law is more explicit in this regard, though perhaps does not address the tradeoffs between social vs. individual benefits, as discussed earlier. De-identification is a kind of legal remedy to this conundrum, but also carries scientific limitations.

Efforts to provide the patient with a more practical means to expand or revoke broader, forward-looking informed consent and authorization processes can: (1) make such consent/authorization more voluntary and use-case specific; (2) allow consent/authorization to be informed by future information not known at the time of the original grant of consent/authorization; and (3) support a weighing of the individual benefits of allowing future research use of case-level clinical or genomic data [36]. Many barriers prevent the participation of underserved populations in clinical research and cancer screening programs. Trust, or the absence thereof, is a determinant of clinical research participation and an essential factor in targeted, effective care earlier in the natural history of disease [41]. Fragmented care is a reality of the healthcare system [42] and can be especially daunting to patients faced with decisions regarding alternative treatment options or participation in a clinical study. There is a disconnect between diagnosis-related data and outcome data. Beyond the temporal disconnect, data boundaries between providers and their organizations make it impractical and often impossible to engage a cancer patient over the entire natural history of disease. This is particularly true during periods before diagnosis (i.e., in cancer screening programs for at-risk populations) and after disease is stabilized (when the intensity of specialist care subsides).

As noted earlier, patient/subject trust is an important determinant of clinical research participation. Although permissible under research rules and ethics, signing an informed consent/authorization document for future research inherently requires a leap of faith for patients. A source of reticence in using consent for future research is that it is difficult and perhaps impossible for an informed consent form or conversation to convey all of the nuances of providing a specimen or accessing health data prospectively. Without consent to participate in a research biobank or database, an opportunity is missed to learn from the study beyond what is gleaned from the interventional protocol. Indeed, in certain circumstances, an IRB can issue waivers of informed consent and authorization within ethical and statutory frameworks, but in a perfect world, this would be a patient-centric activity. Traditional approaches to patient engagement and informed consent/authorization (e.g., static consent) can limit or preclude efforts to expand the engagement of the cohort to further utilize clinical trial data, outcomes, and health utilization for auxiliary studies [43,44]. In fact, clinical trial data have been underutilized beyond the study in which they were collected because of the logistical challenges in consenting and engaging subjects outside the trial window or protocol scope of an interventional study [45]. We argue that there are inherent limitations in a static consent process that cannot transcend the boundaries of the study timeframe, including lost contact, scope expansion or reduction, and the opportunity to collect new data points that might be warranted due to insights from the original protocol. Those who control the data—the patient, the provider, and/or the trial sponsor—may not be readily available to expand consent/authorization or enable auxiliary studies.

The access of practitioners or researchers to these resources has historically been constrained by narrow or ambiguous consent/authorization [46]. The open-ended consent/authorization for the secondary research approach is convenient, but remains somewhat controversial and an approach used judiciously in many environments [47]. Patient and participant attitudes toward future research are nuanced; a spectrum of downstream uses of information and specimens is deemed acceptable, but the broader and vaguer the use, the higher the objection to consent/authorization for future research [48]. The ethical frameworks for using consent/authorization for future research remain far from consensus [46,49]. The trade-off between open-ended and informed consent/authorization is remedied by a dynamic form of consent/authorization. We propose that, in an idealized clinicogenomic registry, the subject would have an opportunity to discuss each subsequent use of their data or specimen, ask questions, and provide truly informed consent. In certain cases involving high complexity, literacy, or language barriers, it may be ideal to have the consent process documented in a video recording. These trust and patient participation issues are amplified in a primary care setting because of the lack of expertise in administering the informed consent process and communicating the importance of future studies. Moreover, rural patients referred to specialty centers may be unfamiliar with the research process, and there may be a lack of cultural competence in the informed consent process. Distance, patient involvement, and lack of family support are also direct impediments to underrepresented minorities’ participation in clinical research, among other issues. Leveraging interprofessional healthcare and the trust center of primary care, as well as empowering the interprofessional teams with connected information systems, has been argued to remedy the distrust underlying vaccine hesitancy in disadvantaged populations [50]. The collection and sharing of genetic data are increasingly prevalent practices in clinical research and drug trials. This raises concerns about historical, ethical transgressions in genetic information. The Genetic Information Nondiscrimination Act of 2008 (GINA) provides some statutory protections and, ostensibly, a baseline for public trust in research involving genetic data, but there are evolving interpretations of the law as new data uses arise [51]. Patient education and cultural competence from healthcare professionals need to be intrinsically linked to increasing the participation of underrepresented patients in clinical research.

Trust is also an issue when randomization to the standard of care or placebo-controlled study arms is possible. Sharing sensitive personal medical information adds a layer of reservation. It is not trivial to engage patients in the standard of care when genetic testing is a theme, so engaging research participants in interventional clinical research within an oncology study amplifies the trust factors impeding cancer care in traditionally underserved and underrepresented populations. Add the perceived risk of unethical use and familial implications of genetic information [52], and it adds up to a significant threshold to engage reticent populations in interventional trials that involve the auxiliary collection of genomics and registry data. This enables prospective participants to be more informed and increases research participation, particularly when genetic information is involved [52]. Obtaining informed consent and authorization is an ethical imperative, but it is also an interpersonal process and is thus laborious. Retaining study participants in clinical research projects is far preferable to the over-recruitment or replacement of subjects lost to attrition—a significant yet indirect and hidden cost of most clinical research studies. Approaching research participant engagement deliberately and thoughtfully is a worthwhile investment. The All of Us program, a national effort to accelerate health research by exploring the relationship between lifestyle, environment, and genetics, acts as a learning laboratory for optimizing many tools and strategies for gathering informed consent in populations with low health literacy [53]. The ideal clinicogenomic registry would allow participants in placebo or control arms to directly benefit from their participation.

### 2.4. Big Data

In oncology, clinically annotated biobanks and registries, and commercial clinicogenomic databases (e.g., Foundation Medicine, Tempus, Flatiron Health, and Guardant) have extended the scope of what can be learned from interventional trials [12]. Layering longitudinal population health and economic outcome studies onto interventional cohorts or synthetic control cohorts remains an opportunity and challenge worthy of attention. Byrd et al. [54] review a number of technological strategies to address the discontinuity of healthcare data that perhaps point to a better future; however, trust will be a crucial determinant of patient participation in these emergent data ecosystems, particularly among underserved and underrepresented populations.

Real-world data (RWD) and real-world evidence (RWE) have emerged as increasingly powerful currency to power knowledge about populations and disease processes unlocked by computational technology and artificial intelligence. Essentially, an ideal registry can enable a clinical trial or a clinical trial arm to be created and analyzed in silico. Several use cases exist already, and the Food and Drug Administration (FDA) has cited a number of them in issuing draft guidance on the use of RWD and RWE:To inform clinical trial design;To support clinical decision support, clinical guidelines, and policy; andTo address post-market safety, adverse events, and regulatory decision making.

Synthetic cohorts [17], external control arms [55], and digital twins [56,57] are providing powerful tools to create clinical trial simulations, understand the clinical trajectory and variability of controls, and even augment and increase the statistical leverage of randomized controlled trials by enabling a much greater understanding of variability and effect size a priori. More recently, examples of drug approvals for registration trials for label expansion [58] in small populations [58] and rare diseases [59] have emerged. Registries play an important role in each of these use cases for RWD and RWE, but the contents of the registries must be aligned with, and be of sufficient quality to build, a clinical trial cohort. For example, in the case of cerliponase alfa for Batten disease, a rare fatal inherited disorder, also known as neuronal ceroid lipofuscinoses, in which the nervous system cannot recycle certain degradation products, the control was disease progression in a historical standard of care cohort [60]. The establishment of a robust and well-curated global registry was crucial in enabling the positive outcome of the cerliponase alfa story [61]. The alignment of patients and other stakeholders around data and specimen sharing can be more complex and challenging in oncology, infectious disease, and chronic diseases. In the nonlethal disease setting, the challenges are magnified by the absence of motivation and urgency, when benefits might be less direct and the relationship between patients and providers less intense. In recent years, some patients and patient advocates have argued for social and economic benefits to accompany data sharing. The ideal clinicogenomic registry would foster transparency and trust among providers, researchers, and patients.

### 2.5. Data Standards

Many practitioners in the oncology and pathology specialties are active in, or familiar with, clinical research and clinical trials. In our experience, for populations who may receive chronic care outside of these specialties, a lack of clinical research expertise and infrastructure at care locations can be a major impediment to recruitment and participation. Assessment of outcomes beyond common oncology endpoints, such as overall survival and progression-free survival, is challenging due to several factors: lack of interest by study sponsors (regulatory relevance), lack of consent, and lack of longitudinal data linkage strategies. For example, in heart failure, the linkage of electronic health record (EHR) data among specialties on the care team is cited as a major determinant of poor outcomes in heart failure management [54]. The long-term benefits of care models for underserved subpopulations remain suboptimally documented [62,63], especially early cancer diagnosis/prevention models [64] and pharmacogenomics [65]. As a result, informed policy decisions addressed disparities (i.e., regulations, reimbursement, and federal Research and Development priorities) suffer. The reimbursement of pharmacogenomic testing has been hindered by a lack of randomized controlled trials (RCT) and RWE of cost-effectiveness [66]. The use of tumor sequencing panels has grown in recent years. Challenges and barriers to adoption and reimbursement remain in the US [67,68], Canada [69], and Europe [70]. The FDA published and regularly updates a data standards catalog which, if followed, ensures registry data have utility in regulatory filings [71]. The ideal clinicogenomic registry would utilize as many established data standards as possible for the corresponding content (i.e., CDISC, HL7, LOINC, and SNOMED). The ideal clinicogenomic registry would meet a quality standard allowing for the creation of simulation cohorts and drug registration trial cohorts and the use of the data in regulatory dossiers.

### 2.6. Boundary Problems

Institutional trust around data stewardship remains a major impediment to data sharing. Protected health information (PHI) is often at the root of the angst. The reality for health systems whose primary line of business is providing high-quality healthcare at a sustainable financial margin is that research and data sharing activities can be perceived as a source of cost and a net liability [72]. Concerns about the loss of control of security and privacy have real and perceived risks to the enterprise, and the benefits of widely sharing these data tend to be more abstract. There are also real and perceived competitive implications of sharing health information, which has resulted in proprietary behavior and, consequently, the limited scaling of query-based health information exchanges [73]. The collection of multiple informed consent documents over the lifecycle of participation can confound or even pose conflicts with respect to the intended scope of consent for different data elements, adding to the data governance concerns of institutions [74]. The sharing of PHI triggers a contract administration (data use agreements) process that is almost always time consuming, fraught with friction, and costly for both investigators and institutions. The result is an aversion to conducting research involving PHI, which can impede some study designs involving re-contact and longitudinal follow-up. Interoperability, security, privacy, and proprietary concerns contribute to the boundary problem that limits the mobility of the health data necessary to successfully link genomics with public health outcomes. This challenge is perhaps most exaggerated in the ambient intelligence application of healthcare, where the near-continuous collection or analysis of personal (and protected) health information is simultaneously transmitted over a multitude of computing platforms, vendors, and organizations [75]. The ideal clinicogenomic registry establishes trusted channels of digital governance and exchange between stakeholders and users, particularly institutions.

### 2.7. Protocols

The battery of data collected in a registry, associated quality control/quality assurance procedures needed for standardized protocol, and data formats vary across registry and clinicogenomic programs. Compiling an apples-to-apples cohort from multiple sources can be difficult to impossible. Google Health and Microsoft HealthVault represent two examples (both now defunct) to provide a stewardship ecosystem for the types of information considered in this review, including SDOH, predicated on consumer-mediated data exchange. It has been argued that a major cause of the failure of these programs was a lack of data standards and interoperability [76]. The ideal clinicogenomic registry would minimize interoperability issues to enable networks to build cohorts from fragmented populations of rare disease phenotypes or genotypes.

Indeed, the Total Cancer Care Protocol implemented by the Moffitt Cancer Center in 2006 reflects this aspiration (ClinicalTrials.gov, accessed on 12 April 2022, Identifier: NCT03977402). The Total Cancer Care Protocol [77] is an example of a unifying approach to data collection and eases the metanalysis of clinicogenomic databases. The Total Cancer Care Protocol aims to standardize data collection across tumor types and address data interoperability. The network of cancer centers participating in the initiative can pool data and build relevant cohorts at a statistically robust scale for testing a clinical hypothesis. For example, assembling a table of allele frequencies in ethnic groups through meta-analysis and cross-referencing of ethnic distributions of pharmacogenomic star alleles in existing databases is not practical (Aponte, Silva, and Ramos, unpublished). This has potential clinical decision-making implications [65] because of the variation in the taxonomies and descriptors used to annotate race, ethnicity, and ancestries.

Genetics is intertwined with social and economic determinants of health. Predisposition toward malignancies, psychiatric afflictions, and metabolic disease are known and often actionable pieces of the health puzzle. For example, genetic factors, a history of liver disease, alcohol and tobacco use, or occupational toxin exposures can warrant lung [78] or liver [79] cancer screening. The collection of potentially useful information like race, ethnicity, and ancestry is not standardized and represents a significant challenge to robustly documenting these factors. Making meaningful comparisons of data collected by different organizations might not be practical [80]. In diverse and admixed populations, as found in the US, ethnicities are not discrete and do not transcend commonly used classifications and nomenclature [81], rendering the race, ethnicity, and ancestry annotation of variant frequencies incomplete, misleading, opaque, and equivocal. Whatever curation standards are used will always be imperfect; however, enabling the comparison of data shared across organizations improves the level of adversity from impossible to challenging. However, knowledge about the distribution and prevalence of known and actionable variants in ethnic minorities trails the knowledge base in populations of European ancestry. This is largely because of the underrepresentation of ethnic minorities in registries and biorepositories, based on deep-rooted distrust and data governance concerns that have arisen from historical, ethical transgressions [82] and the misappropriation of specimens [83]. An ideal clinicogenomic registry gives the patient transparency into the process and deliverables, as well as a role in the governance and downstream use of their information and biospecimens.

A standardized approach to annotating the registry data and specimens collected is necessary. The Patient-Centered Outcomes Research Institute (PCORI) has done much to elevate the inherent trial-and-error nature of medicine from the anecdotal to the systematic. By articulating and funding the validation of standards and best practices, in care, documentation, and dissemination, PCORI has added rigor and intensity to the social considerations of outcomes-based research [84]. Economic factors such as socioeconomic status and healthcare utilization are impactful realities of the systems-nature of health disparities. These factors are seldom robustly factored into registry structures, and opportunities are lost to address the socioeconomic determinants of health disparities. An ideal clinicogenomic registry provides a system’s view into the interplay of socioeconomic factors, genetics, and the natural history of disease.

### 2.8. Technology

Virtual clinical trial models have been proposed and piloted with limited success, as revealed in a recent proceeding organized by the National Academies [85,86]. The digital engagement of participants and patient-reported measures have been acknowledged as readily amenable to virtualization, but recruitment more likely requires elements of the trusted provider relationship. Trusted healthcare relationships are emerging as a hot topic issue in debates about health insurance policy, perhaps signaling a more important role in primary care in chronic disease management and navigating the increasing complexity of the health system. It has been argued that the primary care environment is an environment where weak clinical signals underlying adverse drug reactions can be detected and acted upon preemptively [13]. Our argument is logical in oncology, where malignancy is often a culmination of environmental, social, and biological factors that are computationally tractable and actionable. Payers, such as TriCare [87], and other health systems are actively exploring programs aimed toward value-based care to integrate patient navigation, the longitudinal documentation of care protocols, and healthcare utilization. Such programs would ideally implement rigorous data standards and scientific methodologies to definitively reveal more effective practices that reliably impact individual and population health positively. Texas and California are among a handful of states that have established training standards for healthcare navigators and community health workers (CHW), thus establishing regions where holistic registry programs might be most likely to reliably demonstrate the benefits of CHW, navigators, and novel value-based care models. However, the boundaries and silos in the healthcare ecosystem remain a formidable challenge. Interoperability is a widely recognized challenge in healthcare, value-based care, and research. The Cures Act Final Rule [88] is expected by some to significantly address “engineered interoperability” or the “walled garden effect”, which would ostensibly alleviate one source of friction in moving health data across organizational boundaries.

Using digital tools and virtual clinical research practices to engage cancer patients longitudinally and virtually where they live holds much promise to capture nuance and establish this wider perspective on addressing health disparities across the natural history of disease. Cloud computing, mobile computing, digital ledger technologies, tokenization, and artificial intelligence technologies are powerful tools that could enhance engagement along the data lifecycle. Tokenization, as used here, means converting a valuable piece of data into a form that can be exchanged while preserving security, nonfungibility, and non-exploitability for an intended end user. If properly used, explained, and incentivized, they can bridge the trust barriers necessary to engage underserved populations more substantively in longitudinal clinical research, and link together composite cohorts for data analysis. With accepted best practices and standards for data formats, privacy, and security, such tools can lower the transaction costs and liabilities of data sharing to a feasible threshold, and perhaps enable a currency-like nature of these datasets to help offset the philosophical and financial frictions precluding sharing.

However, there are significant current challenges to the monetization of data. Mikk et al. cite three components for successful patient engagement in the use of their data: (1) moving data across disconnected nodes of the healthcare system; (2) documentation of data from an encounter (a data receipt); and (3) contracts between patients and third parties. The concept of ownership of medical data turns out to be quite opaque, particularly concerning the specific property rights a patient may have [31]. Despite significant efforts to reconcile this issue in the UK, where most health data have a common origin in the National Health Service (NHS), much debate remains over property rights frameworks such as intellectual property [38]. Opportunities for meaningful financial rewards are difficult to envision—the appropriability of valuation and financial flows for medical data is challenging in most real-world instances [32]. Precedent and practice are also barriers to adopting this practice.

A key challenge and opportunity in engaging these populations is that they often engage with the health system at Federally Qualified Health Centers (FQHC) or community health centers. The latter lacks federally mandated electronic health records, and the use of registries in the former has been limited. However, even with limited research on the use of registries in care in FQHCs, there is some promise in diabetes management [89,90]. Technologies that can ensure the provenance and chain-of-custody of digital content and metadata are not widely integrated with administration functions in healthcare or research enterprises, but do warrant attention [91]. For example, biomarker data obtained from subjects on an interventional trial and a federally funded genomic sub-study might require the reconciliation of data rights with multiple third-party contracts. Still, these data flows are not readily available across an enterprise administering data and contractual relationships. Such tools, with accepted best practices and standards for data formats, privacy, and security, can lower the transaction costs and liabilities of data sharing to a feasible threshold, and perhaps enable a currency-like nature of these datasets to help offset the philosophical and financial frictions precluding sharing by providers, health systems, and patients [92]. The ideal clinicogenomic registry provides researchers and institutional stewards of clinical data a direct role in data governance and a means to accrue direct benefits from facilitating registry participation and curating a registry over time.

## 3. Vision for an Idealized Clinicogenomic Registry

In the era of big data and artificial intelligence, demand for RWD to train artificial intelligence models holds much potential to reduce guesswork in clinical trial sizing, hypotheses development, grounding of RCT investments, prioritization of label expansion opportunities, reduction of the number of patients enrolled in RCTs, and the acceleration of evidence-based reimbursement and health policy decision making. Each of these use cases of RWD, no longer constrained by the limits of human intuition, has enormous economic virtue in terms of the potential cost reductions and represents a major catalyst of demand for RWD. However, building a digital cohort using case-level data over time or across organizational boundaries entails myriad ethical, legal, and administrative quandaries, leaving the promise currently out of reach. An idealized tumor registry would provide a systems perspective (with scientific rigor) on the social, economic, genetic, and environmental determinants of health. However, this requires a leap of faith by the patient that is often too high a threshold for socially and economically marginalized populations, who perceive aspects of clinical research participation as subverting their power and autonomy or view the benefits of participation as being too abstract. Indeed, affording both patients and health systems transparency and control over their role in clinical research can be game changing. If marginalized populations are not better represented in clinicogenomic registries, the artificial intelligence tools trained on these assets will continue to amplify health disparities. There is an acute social and moral impetus to reverse the long-standing disengagement of socially and economically marginalized populations from clinical research. Patient-centric data governance promises to give marginalized populations a proactive voice in their participation in clinical research and enable them to directly experience the benefits and consequences of participation.

## 4. Hypothetical Case Study

A person with elevated cancer risk (i.e., a former smoker who qualifies for low-dose computerized tomography (CT) screening for lung cancer) from an underrepresented population (rural African American veteran) is recruited into a US Veterans Affairs Department (VA) lung cancer prevention program. An idealized flow with all of the component parts is represented in Figure 1. At entry, the individual is counseled and provides affirmative consent to share their data relevant to their cancer risks, including ongoing chronic disease management care (chronic obstructive pulmonary disease (COPD)) outside of the oncology specialty, through a nonprofit health information exchange (Cancer Prevention Registry Health Information Exchange (CPRHIE)) with a dynamic, digital research consent layered onto the platform (#1, in Figure 1).

The scope and duration of data collection and distribution are patient-driven and dynamic. At the behest of patients, select healthcare providers, health systems, and payers in the region would cooperate in making available to and through the CPRHIE select EMR data and specimen sets in HL7 coded continuity of care documents for patients (#6 in Figure 1). For research use, an honest broker would tend the health information exchange, and a trusted party would govern sharing. Still, control of an individual’s data use would ultimately be controlled by the individual (#2, #3 and #4 in Figure 1), much in the manner envisioned for the Google Health and Microsoft HealthVault programs. The process for allowing a third party commercial user allows for patient control, consent, and even some form of digital consideration for each commercial use request. That consent/authorization can be directed at all data users (#8, Figure 1), select users (e.g., for-profit companies, health insurers; #9, Figure 1), or none (#10, Figure 1). In the Idealized Clinicogenomic Registry, the person at high risk for lung cancer can participate in the registry before being diagnosed with cancer (#6, Figure 1) and/or after being diagnosed with cancer (#7, Figure 1). The person can again control the use of longitudinal clinical data in the CPRHIE associated with that specimen and its sharing and use. Two years later, the person is diagnosed with early-stage lung cancer, and her tumor is analyzed for mutations. The person is contacted again on a phone app at their home and counseled on potentially expanding data access to participate in a lung cancer health disparities outcome study. The person can provide, expand, terminate, or selectively deny access to individual case-level data elements or all of their data in the CPRHIE using an electronic digital consent agreement (#3, Figure 1). The expanded consent triggers a health record request to the VA to share healthcare utilization data (#7, Figure 1). These data are used to demonstrate the effectiveness of the VA lung cancer screening program to improve prevention and survival, and to convince the Centers for Medicare Services (CMS) to reimburse low-dose CT screening. Six years later, the person receives a text that a health insurance company wishes to use the person’s data in a study of patients who have recovered from early-stage lung cancers to expand coverage for a new drug to prevent lung cancer (#9, Figure 1). Ten years after entering the registry, the person opts out of consent to share those data by sliding left on a mobile app (#5, Figure 1). The case-level data from that person would not show up in future queries by users of the external portal of the idealized clinicogenomic database.

Distributed ledger technologies [93] would readily enable the envisioned level of connectivity [85] and a tamper-resistant and tamper-evident record of the provenance and annotation of the CPRHIE and tumor registry datasets into a longitudinal data rendering of that patient journey (case level vignettes). This ledger is generated automatically by data creation and updates during the data or patient journey and all data transactions. At the same time, the granular case-level data from the CPRHIE would readily enable deterministic, probabilistic, and referential matching [54] to case-level data from databases like the All-Payer Claims Database, further expanding the profile of the clinical case, but only if the patient/participant is amenable. The research participant, the human subjects protection officer, the IRB, the privacy officer, and the legal office would all have the digital control to liberate or restrict the access or egress of data on such a system in accordance with their roles, preferences, or obligations (i.e., contracts and consent).

Blockchain The integration of longitudinal case-level data from numerous providers in the patient journey is so complex that it is generally impractical under current practices in the healthcare industry [16]. A coalition of academic and industry data holders has published a report demonstrating the promise of this approach in a large lung cancer cohort assembled from real-world (medical records from 275 oncology clinics) and commercially curated data [94]. Data abstracted from medical records included: smoking status, date of advanced disease diagnosis, biomarker status, and dates of disease progression. The study also utilized derived endpoints, including overall survival, time receiving therapy, maximal therapeutic response, and clinical benefit rate. This project involved a relatively large cohort of over 4000 lung cancer cases. This project could be conducted primarily because of the carefully abstracted and curated medical records of Flatiron Health using case-level data that were deidentified through a rather laborious process. Because the data sources of this program were limited to medical records from the oncology silo of the healthcare system, further longitudinal examination of the cases with RWD data from outside of the oncology specialty was likely impractical. In short, as impressive and insightful as this clinicogenomic study proved to be, medical and SDOH from before or after the cancer treatment window could not be evaluated; such is the challenge with most cancer registries. New tools are warranted to enable the management and flow of longitudinal clinical data across the natural history of disease and the organizational boundaries of that patient journey.

Blockchain is a versatile technology with many attributes useful in addressing the challenges confronting the idealized clinicogenomic registry. Table 1 illustrates the benefits of a clinicogenomic registry, common challenges, and blockchain solutions. Inherently, blockchain technology arose as a solution to a fundamental problem: converting any kind of data, including health data, into an asset and maintaining a digital ledger of that data element’s journey through time. The impedance of the movement of health data contributes to the disparate representation of underrepresented minorities and other disadvantaged populations in clinicogenomic studies; this translates to bias in training artificial intelligence models. That sounds similar to many of the liquidity and mobility challenges facing clinical data in the healthcare ecosystem. The capacity to maintain and accumulate metadata (e.g., consent permissions, who has downloaded, what data elements were shared) around primary data or a specimen makes that data nonfungible [95]. A corollary would be a tracking number and all digital information about timestamps and chain of custody for a package throughout a geographic supply chain. A robust, private, permissioned blockchain platform that enables the collection of health data in normal life, health, and disease can add a virtual and longitudinal dimension to clinical research by annotating the data journey and providing the patient with visibility and agency during that journey. Such an approach also provides incentives (population-scale outcome data) for the necessary cooperation of select participating health systems, where data from relevant non-oncology care or preventative care (e.g., primary care, ob-gyn) might help illuminate the cancer care viewpoint. A clinicogenomic database architecture that gives providers and patients/participants visibility and agency into data sharing lowers the high activation energy that has impeded effective participation of these two stakeholder groups in health information exchanges.

Mackey et al. present a set of “fit-for-purpose blockchain” considerations for the management of healthcare data [93]. These include the following.

The **governance** capabilities of a private blockchain enable stakeholders and communities of users to dynamically control the permissioning and consent mechanisms for how data are used. This means that contract officers, privacy officers, and honest brokers can assume direct digital control of data sharing in accordance with legal contracts, informed consent documents, protocols, institutional policies, and statutory constraints. Digital governance also makes the generation of granular data sharing possible in near-real time by all stakeholders (providers, researchers, and administrators), but with individual-level control. This is more efficient than identifying a mutually agreeable honest broker or intermediary in the clinic or institutional administration to perform this function at a bulk data level. Interestingly, blockchain-based governance tools can be used to digitally enforce de-identification, bringing both transparency and control to privacy officers, while potentially making the de-identification of data a less manual process. Notably, the smart contract and governance layers can make it practical to provide agency directly to patients for their permitted uses of their health data, thus creating a practical, patient-centric approach to data management.

The **digital ledger** capabilities enable any patient, auditor, or other interested party to examine a transparent ledger of who received or used a particular dataset or specimen. Most blockchain architectures render these ledgers tamper-evident and tamper-resistant, ensuring the provenance of consent for a specific specimen or patient record. This living-ledger functionality is useful for audits in general, and HIPAA compliance reporting specifically. By extension, such information could readily be used as the basis for micropayments to compensate and incentivize sharing at the patient or case level, and perhaps inform pricing and market-making mechanisms for case-level data [32]. For example, suppose a pharma company pays $100,000 to use a finite set of data elements to train an artificial intelligence model. In that case, the ledger enables the digital accounting and sharing of revenues, with stakeholders providing access to each case-level element, individual, and institution. The ledger, metadata, and audit trail effectively make a data element trusted data [32]. Much like a DNA barcode carries metadata in a bioanalytic assay, a blockchain ledger can offer the same for a data element. This is the basis for blockchain technology in the nonfungible token market [96].

The **smart contracting** capabilities of blockchain can provide a patient—or any official in a specimen/data supply chain with a handheld computer—the capability of permitting or denying access to a registry resource, whether an entire EHR or a single data object, which can be limited to a single genotype for a single allele for a single patient. A user interface can be designed to utilize and implement unlimited data governance and annotation functions that are patient-facing, provider-facing, administrator-facing, or all of the above. This allows any stakeholder to expand or revoke consent with the ease of a mobile app, much like a click-through license for a piece of software or a downloaded app. Smart contracting can also lower transaction costs by parallelizing the functions of intermediaries (i.e., compliance officer, privacy officer, provider) and digitizing the myriad administrative workflows associated with institutional consensus-building, while providing regulatory, ethical, and administrative stakeholders with direct and granular control of subject data.

The **distributed** nature of the blockchain means there is no single failure point from which to access (or destroy) large troves of clinical data. Further, the underlying data are often securely distributed on multiple storage locations rather than on a single server in a single location. This feature can reduce the imputed and ongoing cost of the sharing, ownership, or possession of clinical data and PHI in collaborative research—converting what is arguably an administrative liability into an operational asset. Further, a distributed network of highly annotated health data with metadata reflecting provenance, ethical review, and granular consent are readily amenable to moving select data to edge computing nodes where fit-for-purpose clinical data warehouses can be constructed for research projects and analytics collaborations. This feature addresses a major concern of health systems: the compromise of perceived competitive and proprietary advantages in allowing the mining of larger datasets.

The **monetization** of data using blockchains has the potential to securely unlock data as a form of currency, and better align incentives for sharing among stakeholders (patients, health systems, healthcare companies) with benefits that are important to the individual stakeholder [32]. While abstract and beyond the scope of this piece, and capably reviewed elsewhere [97], generally speaking, blockchain-based data marketplaces leverage digital exchanges to digitize the value of assets ranging from medical data to algorithms. Consequently, the monetization of health data can democratize genomic data ownership (and the associated annotation) while actively engaging patients and institutional stakeholders (i.e., compliance officers and privacy officers) in ensuring the provenance of research in near-real time.

## 5. Conclusions

The mosaic of walled gardens within the medical informatics ecosystem makes transparency and trust difficult across time and organizations. The longitudinal collection of medical data across providers, labs, and payers remains a barrier to value-based care, but is also a barrier to understanding and addressing the social, economic, environmental, and biologic determinants of health and health disparities. Continuity of care is especially fragmented in impoverished and socially marginalized populations, further amplifying the challenges in understanding the interplay between their intrinsic biologic and social determinants of health. The legitimate privacy and security concerns (lack of trust) of stakeholders about the stewardship of big datasets in healthcare are especially acute in these participant populations. The remedy is the flexibility and control to change one’s mind and opt out.

An intractable number of administrative intermediaries impede data flow across organizational boundaries and through the natural history of disease. We argue that technologies that bring trust and transparency to virtualize and disintermediate clinical research can potentially lower the activation energy to choose participation for otherwise skeptical populations experiencing health disparities. Empowering patients and their providers to manage their health, incentivizing their engagement, and giving them visibility and control of their data are necessary for building trust. Artificial intelligence technologies are creating an increasing gravitational pull on longitudinal datasets, and market mechanisms are needed to advance the equitable delivery of healthcare. Artificial intelligence technologies have also demonstrated the risk of amplifying health disparities and biases rooted in the Preferred Cohort effect and reclusion from clinical research. Establishing a registry in a data ecosystem operated by an honest broker, with tools to provide subjects/patients and institutional stakeholders with visibility on the provenance on the chain of custody of every element of data (or biospecimen) that goes into it, would be a paradigm shift. The idealized clinicogenomic registry would bring a patient-centered application of technology to reduce the administrative and ethical challenges that impede clinical data sharing.

## Figures and Tables

**Figure 1 jpm-12-00713-f001:**
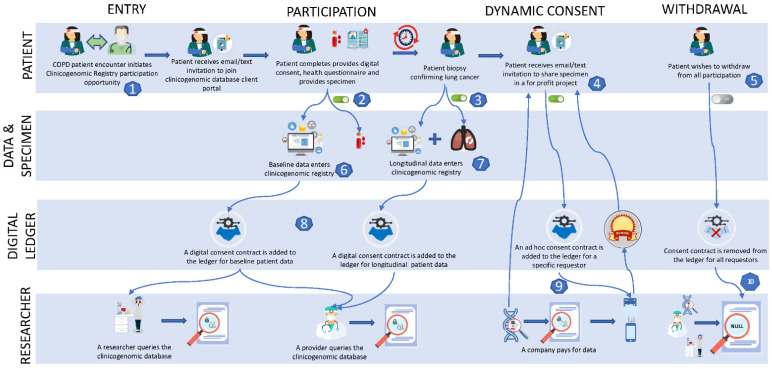
Schematic diagram of the patient and data journey in an idealized clinicogenomic database administered on a digital-blockchain ledger.

**Table 1 jpm-12-00713-t001:** Idealized clinicogenomic registry features, challenges to complementation, and features of blockchain technology addressing these challenges.

Ideal Registry	Challenge	Blockchain Feature to Solve It
Patients provide consent for a wide swath of research activities	Patient control of future use of data	Governance; smart contracts
Incentives for health systems and patients to share data	The chain of custody of fungible data makes attribution of bulk data impractical	Monetization; smart contracts; digital ledger
Users and providers have comfort with provenance of data in a collaboration	Third-party obligations and lack of granularity of data and specimens shared	Digital ledger; governance; smart contracts
Assembling a cohort involves minimal institutional touchpoints and bypasses cumbersome processes	Transactional frictions of health data sharing	Governance layers; smart contracts
Complete control of who uses data and for what purpose	Unauthorized use or replication of fungible data	Smart contracts; digital ledger
Low-friction methods to define the rules of engagement for compliance and legal constraints for data recipients	Operational costs to administer data governance and administer legal contracts	Governance layers; smart contracts
Facile HIPAA and compliance reporting	Lack of granularity of bulk data and lack of visibility to compliance administrators	Smart contracts; digital ledger

## Data Availability

Not applicable.

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
