# Peer review of "An Idealized Clinicogenomic Registry to Engage Underrepresented Populations Using Innovative Technology"

_jpm, 2022, doi:10.3390/jpm12050713_

Round 1
Reviewer 1 Report
This manuscript is well framed and written, I don't have major comments, this work can be accepted after minor revisions (line 7 et al).
Author Response
Dear Reviewer- Please see attached responses to review.

Reviewer 2 Report
This manuscript has been written in a very confusing way, and extensive editing is needed as follows:
- There is a problem in using abbreviations throughout the manuscript. The full term should be mentioned first with the abbreviation between paresis then the abbreviations should be used throughout the manuscript. E.g., In line 79, “social determinants of health” has been abbreviated as SDOH then the full term was repeated in lines 351 and 560. Such errors have been repeated for most abbreviations throughout the manuscript. Also, in line 45, the full term of BRCA should be first mentioned, then the abbreviation used further.
- The title is very long and needs to be rewritten.
- Where are the keywords?
- The abstract needs to be rewritten, focusing on the topic rather than general information.
- The sections of the manuscript are overlapped and need to be reorganized.
- Figure 1 needs to be reconstructed as the details are unclear in their present form.
- Table 1: wrong formatting.
Author Response
Dear Reviewer 2: Please see attached responses to review.

Reviewer 3 Report
I would like to thank the Authors of the Manuscript “An idealized clinicogenomic registry to engage underrepresented populations: innovations in technology: enhancing public health through innovative approaches and technologies” for the opportunity to provide commentary on their work.
As a general note, I find the Manuscript very engaging and the content is quite clearly presented, even if there are almost 15 pages of full text, packed with information. The overall structure has thrown me off a little, as this is not the canonical partitioning of an Original Research article, but it appears to me as a hybrid between an extended Review, a long Commentary piece and a well-executed theoretical exercise. I would like to point out that this is not detracting anything from the overall quality of the work presented here; indeed, it provides the Reviewer with an instrument to critique its own ways of dealing with the textual material and its contents.
You will find my suggestions for improvement in the following list.
1) Change the title to a more synthetic one (20 words should be a good upper limit); I would strongly advise against the use of two columns in the same sentence.
2) The whole manuscript is rich in punctuation errors and double spaces at the start of many sentences; please, make sure to re-read and check for these mistakes.
3) lines 30-33: “molecular characterization of disease” is used twice in the same sentence
4) lines 30-34: these sentences do not have a reference. Although I understand that examples are provided later, I suggest adding a couple of references here to highlight the challenges (e.g. Taube, Sheila E et al. “A perspective on challenges and issues in biomarker development and drug and biomarker codevelopment.” Journal of the National Cancer Institute vol. 101,21 (2009): 1453-63. doi:10.1093/jnci/djp334
5) lines 51-53: this sentence is not clear in its meaning, because the structure seems wrong.
6) lines 80-86: I would suggest adding some reference here, as there are plenty of studies on the topic of rural and marginal healthcare avoidance based on trust (Feng, Yingchao et al. “Effects of the Two-Dimensional Structure of Trust on Patient Adherence to Medication and Non-pharmaceutical Treatment: A Cross-Sectional Study of Rural Patients With Essential Hypertension in China.” Frontiers in public health vol. 10 818426. 4 Mar. 2022, doi:10.3389/fpubh.2022.818426; Chen, Wenqin et al. “Effect of trust in primary care physicians on patient satisfaction: a cross-sectional study among patients with hypertension in rural China.” BMC family practice vol. 21,1 196. 21 Sep. 2020, doi:10.1186/s12875-020-01268-w; Wang, Mengxiao et al. “The Associations Between Sociodemographic Characteristics and Trust in Physician With Immunization Service Use in U.S. Chinese Older Adults.” Research on aging vol. 44,2 (2022): 164-173. doi:10.1177/01640275211011048; Cao, Qiuchang Katy et al. “Trust in physicians, health insurance, and health care utilization among Chinese older immigrants.” Ethnicity & health, 1-18. 18 Jan. 2022, doi:10.1080/13557858.2022.2027881)
7) lines 141-142: HSPO is unnecessarily spelled out twice
8) lines 152-155: this sentence needs punctuation, otherwise it is hardly intelligible
9) lines 161-162: what are the Belmont Reports, how do you find them, what is their applicability (regional, national [US only?], supranational) and can you add a reference or a website to reach them?
10) line 163: what does HIPAA stands for, how do you find it, what is the applicability (regional, national [US only?], supranational) and can you add a reference or a website to reach it?
11) since the article has a somewhat unusual structure, there are whole paragraphs in which it is not clear if there is need for references or the reasoning is presented as an original contribution provided by the Authors. I would advise, in any case, to add references when in doubt, and to neatly clarify the outstanding contributions of the Authors throughout the Manuscript.
Author Response
Dear Reviewer- Please see attached responses to reviews.

Round 2
Reviewer 2 Report
No further comments to be addressed
Reviewer 3 Report
I would like to thank the Authors and Editors for the opportunity to comment on the revised version of this Manuscript.
I really appreciate that all my concerns have been properly addressed and relevant changes have been carried out in the structure of the article.
Overall, I feel satisfied that this text can be fully appreciated by the readers for its interesting point of view and the approach used by the Authors on such complex topic.